# Spilanthol from Traditionally Used *Spilanthes acmella* Enhances AMPK and Ameliorates Obesity in Mice Fed High-Fat Diet

**DOI:** 10.3390/nu11050991

**Published:** 2019-04-30

**Authors:** Wen-Chung Huang, Hui-Ling Peng, Sindy Hu, Shu-Ju Wu

**Affiliations:** 1Graduate Institute of Health Industry Technology, Research Center for Food and Cosmetic Safety, College of Human Ecology, Chang Gung University of Science and Technology, Taoyuan City 33303, Taiwan; wchuang@mail.cgust.edu.tw (W.-C.H.); hlpeng@mail.cgust.edu.tw (H.-L.P.); 2Division of Allergy, Asthma, and Rheumatology, Department of Pediatrics, Chang Gung Memorial Hospital, Linkou, Taoyuan City 33303, Taiwan; 3Department of Cosmetic Science, College of Human Ecology, Chang Gung University of Science and Technology, Guishan Dist., Taoyuan City 33303, Taiwan; sindyhu@hotmail.com; 4Department of Dermatology, Aesthetic Medical Center, Chang Gung Memorial Hospital, Linkou, Taoyuan City 33303, Taiwan; 5Department of Nutrition and Health Sciences, Research Center for Chinese Herbal Medicine, College of Human Ecology, Chang Gung University of Science and Technology, Taoyuan City 33303, Taiwan

**Keywords:** spilanthol, 3T3-L1 cells, anti-obesity, AMPK, adipogenesis, lipogenesis

## Abstract

Spilanthol (SP) is a bioactive compound found in *Spilanthes acmella*, giving the flowers and leaves a spicy taste. Studies found that phyto-ingredients stored in spice plants act against obesity-related diseases. SP has antimicrobial, anti-inflammatory, and analgesic properties, but the effects on obesity are not yet known. We investigated the effects of SP in differentiated adipocytes (3T3-L1 cells) and mice fed a high-fat diet (HFD). SP significantly inhibited intracellular lipid accumulation and significantly reduced the expression of lipogenesis-related proteins, including acetyl-CoA carboxylase (ACC) and fatty-acid synthase (FAS). In contrast, SP increased the expression of carnitine palmitoyltransferase (CPT)1 and AMP-activated protein kinase (AMPK) in adipocytes. However, SP suppressed the levels of cyclooxygenase-2 (COX-2), phospho-p38 (pp38), and phospho-JNK (c-Jun N-terminal kinase) (pJNK) in LPS (lipopolysaccharide)-stimulated murine pre-adipocytes. SP administered to HFD-induced obese mice via intraperitoneal injections twice a week for 10 weeks decreased body weight gain, visceral adipose tissue weight, and adipocyte size. SP inhibited lipogenic proteins FAS and ACC, and suppressed adipogenic transcription factors, enhancing lipolysis and AMPK protein expression in the liver. SP has anti-obesity effects, upregulating AMPK to attenuate lipogenic and adipogenic transcription factors.

## 1. Introduction

Obesity is caused by excessive triglyceride (TG) accumulation, primarily due to imbalanced energy homeostasis. Obesity is one of the most prevalent diseases worldwide [1] and causes chronic inflammatory reactions, which activate T-cell and macrophage infiltration into adipose tissues, resulting in the release of inflammatory mediators, such as tumor necrosis factor (TNF)-α, interleukin (IL)-1β, and monocyte chemoattractant protein (MCP)-1, from adipose tissue [2]. Importantly, insulin resistance is induced by obesity–leptin resistance, which activates the inflammatory mitogen-activated protein kinase (MAPK) pathway [3]. In addition, obesity-induced cytokines, especially MCP-1, promote anti-inflammatory M2 macrophage transformation into inflammatory M1 macrophages with increasing obesity-induced inflammation and the expression of a variety of proteins, such as cyclooxygenase-2 (COX-2) [4]. Obesity-induced inflammation is associated with the development of insulin resistance, type 2 diabetes, hypertension, atherothrombosis, and chronic inflammatory diseases [5,6]. Obesity is a complex multifactorial disease, but novel therapeutic dietary interventions may have anti-inflammatory components that are beneficial.

Adipocytes are the depot for energy storage, but excessive numbers and size result in obesity. Pre-adipocyte differentiation into adipocytes is regulated by the transcription factors CCAAT/enhancer-binding protein (C/EBP), peroxisome proliferator-activated receptor (PPAR), and sterol regulatory element-binding protein (SREBP). Increasing C/EBPα expression can prompt 3T3-L1 adipocyte differentiation, and PPARγ activates numerous genes in adipogenesis [7,8]. In addition, acetyl-CoA carboxylase (ACC) catalyzes acetyl-CoA to malonyl-CoA, and then malonyl-CoA is synthesized into fatty acid (FA) by fatty-acid synthase (FAS) during triglyceride production as lipogenesis [9]. In lipolysis, triglyceride lipase (ATGL) initiates TG hydrolysis in adipocytes [10]. Thus, blocking the expression of lipogenic and adipogenic transcription factors and promoting the activity of lipolysis proteins may reduce severe obesity. Studies found that sirtuin 1 (SIRT1)-mediated AMP-activated protein kinase (AMPK) activation regulates the cell energy metabolism involved in lipolysis [11]. Activation of SIRT1 and AMPK signaling decreases the expression of lipogenic enzymes and downregulates transcription factors involved in adipogenesis, respectively. Interestingly, SIRT1–AMPK increases the rate of FA oxidation [12,13,14]. Therefore, enhanced hepatic SIRT1–AMPK signaling may decrease obesity.

Spilanthol (*N*-isobutyl-2*E*, 6*Z*, 8*E*-decatrienamide) is found in the flowers, leaves, and roots of *Spilanthes acmella* as the major bioactive compound of this species. *S. acmella* has a wide range of medicinal effects, including hemostatic, analgesic, and diuretic properties, and is used to treat psoriasis and toothache in India [15]. In addition, *S. acmella* is used for the treatment of toothache in folk medicine in Taiwan and other areas of east Asia. Studies reported that *S. acmella* extract has anti-inflammatory, analgesic, anti-oxidant, antibacterial, immunomodulatory, and antifungal activity [16,17,18,19,20]. *S. acmella* has a pungent taste and causes a numbing and tingling sensation when people touch the flowers and leaves. When the plants are cooked, they lose the strong taste and flavor and can be edible as a green leafy vegetable. Fresh *S. acmella* leaves can also be used as a spice, similar to chili and garlic, to increase the flavor of salads, or cooked leaves can be used in stews and soups [21,22]. Interestingly, *S. acmella* used as a spice results in an estimated average daily intake of 24 μg of spilanthol/person/day in the European Union [23,24]. Studies indicate that many spices (e.g., curcumin in turmeric, capsaicin in red chili, and piperine in black pepper extracts) have antioxidant and anti-inflammatory activities against obesity and attenuate type 2 diabetes, atherosclerosis, and other obesity-related metabolic diseases [25,26,27,28].

The anti-obesity effects of spilanthol are not known. In the present study, we investigated whether spilanthol modulates lipogenesis and adipogenesis in differentiated adipocytes and improves the metabolic profile of visceral adipose tissue in high-fat diet (HFD)-induced obese mice (Figure 1A). We also evaluated whether spilanthol reduces the inflammatory response in pre-adipocytes.

## 2. Materials and Methods

### 2.1. Preparation of Spilanthol and Cell Culture

Spilanthol [29] (ChromaDex, Irvine, CA, USA; Figure 1B) was dissolved in dimethyl sulfoxide (DMSO) and prepared as a 100 mM stock solution that was stored at −20 °C. In culture medium, the final DMSO concentration was ≤0.1%, as described previously [9]. Mouse 3T3-L1 pre-adipocytes were purchased from Bioresource Collection and Research Center (BCRC, Taiwan) and cultured in Dulbecco’s modified Eagle medium (DMEM) (Invitrogen-Gibco, Paisley, Scotland) containing 10% heat-inactivated calf serum (Invitrogen-Gibco) at 37 °C in a humidified atmosphere of 5% CO_2_. 3T3-L1 pre-adipocytes (5 × 10^4^) were pretreated with or without various concentrations of spilanthol (3–100 μM) for 1 h, and then 1 μg/mL LPS (Lipopolysaccharide) added for 24 h. Subsequently, 3T3-L1 pre-adipocytes were lysed for Western blot analysis.

### 2.2. Adipocyte Differentiation

3T3-L1 pre-adipocytes were cultured in DMEM containing 10% fetal bovine serum (FBS) and induced with 1 μM dexamethasone, 0.5 mM 1-isobutyl-3-methylxanthine, and 10 μg/mL insulin for two days. The medium was then exchanged for DMEM medium containing 10 μg/mL insulin for two days, and the DMEM medium was replaced every two days until day 8. The differentiated adipocytes were treated with or without various concentrations of spilanthol (3–100 μM) on day 8 for 24 h, and the 3T3-L1 adipocytes were lysed for Western blot analysis.

### 2.3. Animals and Spilanthol Administration

Four-week-old C57BL/6 male mice were purchased from the National Laboratory Animal Center in Taiwan and housed in polycarbonate cages under constant conditions (12-h dark/light cycles, room temperature 21 ± 2 °C, and humidity 45–65%). All animal experiments were approved by the ethical guidelines set out by the Laboratory Animal Care Committee of Chang Gung University of Science and Technology (IACUC Approval Number: 2016007). After acclimating the animals for seven days, the mice were randomly divided into four groups and treated for 16 weeks: (1) normal diet (ND, *n* = 8), the mice were fed 11% fat normal diet and received DMSO via intraperitoneal injection (ip); (2) high-fat diet (60% energy from fat; HFD, *n* = 8), the mice were fed 60% fat normal diet and received ip DMSO; (3) HFD+SP5 (*n* = 8), the mice were fed a HFD and received ip 5 mg/kg spilanthol dissolved in DMSO; (4) HFD+SP10 (*n* = 8), the mice were fed a HFD and received ip 10 mg/kg spilanthol dissolved in DMSO. All of the ip injections were performed twice a week for 10 weeks. HFD (D12492) was purchased from Research Diets, Inc (Middlesex County, New Jersey, United States). At the end of the experimental period, all animals were fasted for 12 h and sacrificed. Blood samples were collected and the visceral adipocyte tissue and liver removed and weighed after rapidly rinsing with normal saline solution. The samples were stored at −80 °C.

### 2.4. Oil Red O Staining

3T3-L1 cells were treated with or without SP (3–100 μM) for 24 h in six-well plates and fixed in 10% formalin for 30 min. The cellular lipid content was stained with Oil Red O dye solution (Sigma Chemical, St. Louis, MO, USA) at room temperature for 1 h. The plates were washed with phosphate-buffered saline (PBS) three times and 100% isopropanol was added. The intracellular lipid accumulation was photographed using an optical microscope (Olympus, Tokyo, Japan).

### 2.5. Fluorescence Staining

To analyze 3T3-L1 cells, oil droplets and lipid peroxidation were stained with BODIPY^®^493/503 and BODIPY^®^581/591 C1 (Invitrogen, Carlsbad, CA, USA), respectively. The 3T3-L1 cells were treated with or without SP (3–100 μM) for 24 h in six-well plates. After suctioning out the medium, the cells were washed with PBS and fixed in 10% formalin for 3 h. After washing with PBS, the fluorescent dyeing agent was added (BODIPY493/503 or BODIPY581/591). The dye was washed out with PBS and 4′,6-diamidino-2-phenylindole (DAPI) was added to stain the nucleus. A fluorescence microscope (Olympus, Tokyo, Japan) was used to observe the intracellular lipid accumulation and lipid peroxidation.

### 2.6. Biochemical Analysis

Blood samples were centrifuged at 6000 rpm for 5 min at 4 °C and the collected serum was used for analysis of the leptin and lipid concentrations, including triglycerides (TGs), total cholesterol (TC), and high-density lipoprotein cholesterol (HDL-C). Leptin levels were determined using a commercially available enzymatic reagent kit (R&D Systems, Minneapolis, MN, USA). The leptin levels were determined using a microplate reader (Multiskan FC, Thermo Fisher Scientific, Waltham, MA, USA) and measuring the absorbance at 450 nm. Serum TG, TC, and HDL-C levels were analyzed by commercially available TG-P III, TCHO-P III, and HDL-C-P III D FUJI slides in a FUJI DRI-CHEM analyzer according to the manufacturer’s instructions (Fujifilm, Co., Tokyo, Japan).

### 2.7. Histopathological Examination

After the mice were sacrificed, the epididymal adipose, inguinal adipose, and liver tissues were collected and weighed. All of the tissues were fixed in 10% paraformaldehyde, embedded in paraffin wax, and sectioned into 5-μm sections. The epididymal and inguinal adipose and liver sections were stained with hematoxylin and eosin (H&E) and observed using an optical microscope (Olympus, Tokyo, Japan).

### 2.8. Western Blot Analysis

After treatment with spilanthol, the cells were lysed in protein lysis buffer (Sigma, St. Louis, MO, USA). Protein samples (10–30 μg) were separated on 10% SDS polyacrylamide gels and transferred to polyvinylidene fluoride membranes (PVDF; Millipore, Billerica, MA, USA). The PVDF membranes were incubated with primary antibodies, including β-actin (Sigma, St. Louis, MO, USA); COX-2, heme oxygenase-1 (HO-1), AMPK (Santa Cruz, Delaware Avenue, Santa Cruz, CA, USA); JNK, phospho-JNK (pJNK), p38, phospho-p38 (pp38), CPT1, FAS, SREBP-1, SIRT1 (Cell Signaling Technology, Danvers, MA, USA); ATGL, ACC, phospho-ACC (pACC), PPARα, PPARγ, C/EBPα, or C/EBPβ (Abcam, Cambridge, MA, USA), overnight at 4 °C. Tris-buffered saline with Tween-20 (TBST) buffer was used to wash the membranes three times, and the membranes were incubated with secondary antibodies at room temperature for 1 h. Luminol/enhancer solution (Millipore, Billerica, MA, USA) was used to detect the signals and the BioSpectrum 600 system (UVP, Upland, CA, USA) was used to quantitate protein bands.

### 2.9. Statistical Analysis

One-way analysis of variance (ANOVA) and Tukey’s test were used to assess the values. The data were presented as the means ± standard deviation (SD) with *p* < 0.05 considered significant.

## 3. Results

### 3.1. Spilanthol Inhibits Lipid Accumulation in 3T3-L1 Cells

Spilanthol reduced lipid droplet accumulation in 3T3-L1 adipocytes compared to control differentiated adipocytes in a dose-dependent manner (Figure 2A). The use of BODIPY^®^493/503 confirmed the significant decrease in lipid accumulation and lipid peroxidation with spilanthol (Figure 2B). The MTT (3-(4,5-Dimethylthiazol-2-yl)-2,5-diphenyltetrazoliumbromide) assay was used to evaluate the cytotoxicity of spilanthol in 3T3-L1 adipocytes. Spilanthol at concentrations ≥100 μM showed no significant cytotoxicity in 3T3-L1 cells (data not shown).

### 3.2. Spilanthol Regulates Lipogenic Pathway Enzymes in 3T3-L1 Cells

To elucidate the mechanisms underlying the lipogenic effects of spilanthol, we analyzed the phosphorylation of ACC (pACC), as well as ACC and FAS protein expression. Spilanthol significantly increased the ratio of pACC and ACC protein expression, with lower FAS protein expression compared to control differentiated adipocytes (Figure 3A). Previous studies indicated that AMPK regulates energy metabolism via downregulation of the expression of lipogenic proteins (ACC and FAS) and activation of CPT1, with activated CPT1 transporting FAs into the mitochondria for β-oxidation [30]. We evaluated whether spilanthol enhances AMPK and CPT1 expression in 3T3-L1 cells and found that the protein expression of both was significantly increased in the presence of spilanthol compared to control differentiated adipocytes (Figure 3B,C). These data suggest that spilanthol is dependent on AMPK for inhibiting lipogenesis and involves controlling free FA β-oxidation during the differentiation of 3T3-L1 cells.

### 3.3. Spilanthol Reduces the Inflammatory Response by Downregulating MAPK Signaling Pathways in 3T3-L1 Pre-Adipocytes

Previous studies reported inflammatory responses in adipose tissue by pre-adipocytes, which share numerous phenotypic features with macrophages and have a greater inflammatory response than mature adipocytes mediated by MAPK signaling [31]. Obesity-induced chronic low-grade inflammation is associated with the development of a variety of metabolic diseases [28]. Therefore, we investigated whether spilanthol reduces the inflammatory response in 3T3-L1 pre-adipocytes. In the present study, spilanthol significantly suppressed inflammatory mediator COX-2 and promoted anti-inflammatory protein HO-1 expression (Figure 4A). In addition, spilanthol 3 μM significantly suppressed the phosphorylation of JNK and p38 (Figure 4B,C) but did not inhibit the phosphorylation of ERK1/2 (Extracellular signal-regulated kinase 1/2) (data not shown). Taken together, our data indicate that spilanthol may be beneficial for preventing pre-adipocyte inflammatory responses to reduce obesity-related metabolic disease.

### 3.4. Spilanthol Ameliorates Adiposity and Visceral Adipocyte Tissue in HFD-Induced Obese Mice

Spilanthol treatment significantly ameliorated HFD-induced obesity and weight gain compared to HFD-fed mice. We found no difference in total calorie intake (kcal) per day among all groups, suggesting that spilanthol did not reduce the calorie intake to reduce body weight gain (Figure 5). To clarify the suppression of visceral adipocyte tissue by spilanthol to ameliorate weight gain, we measured epididymal and inguinal adipose tissue depot masses. Epididymal adipose tissue and inguinal adipose tissue depot masses were lower in spilanthol-treated HFD-fed mice than in HFD-fed control mice. In addition, H&E staining showed that spilanthol-treated HFD-fed mice had significantly smaller adipocytes in the epididymal and inguinal adipose tissue compared to HFD-induced obese mice (Figure 6). The findings suggest that spilanthol decreased the visceral adipocyte tissue and ameliorated obesity in HFD-induced obese mice.

### 3.5. Spilanthol Ameliorates Hepatic Lipid Accumulation, Improves TC and HDL-c Levels, and Attenuates Serum Leptin in HFD-Induced Obese Mice

Studies demonstrated obesity-induced nonalcoholic fatty liver disease (NAFLD) in HFD-fed mice [32]. Therefore, we evaluated the effect of spilanthol on liver lipid accumulation in obese mice fed an HFD. We found no significant difference in liver weight among the groups (Figure 7A). However, H&E staining of the liver showed that spilanthol-treated HFD-fed mice had significantly fewer hepatic lipid droplets than HFD-induced obese mice (Figure 7B).

Higher serum concentrations of metabolites are associated with lipid homeostasis in HFD-fed mice [33]. We found that the HFD group treated with 10 mg/kg spilanthol had lower serum TC levels and higher HDL-C levels than HFD-fed control mice. However, serum TG concentrations were not significantly different among the groups (Figure 7C–E).

Obesity is frequently associated with high serum levels of leptin. Leptin is secreted from adipocytes to maintain body weight and energy homeostasis [34]. Studies indicated that leptin in the hypothalamus mediates the anti-obesity actions and activation of the sympathetic nervous system (SNS) to induce lipolysis in white adipose tissue and thermogenesis in brown adipose tissue [35]. Serum leptin level is proportional to the adipocytes; hence, we analyzed whether spilanthol improves obesity and decreases leptin levels. In addition, studies showed that obesity-induced leptin and insulin resistance are associated with impaired hypothalamic regulation of energy homeostasis [35,36]. Serum leptin levels were significantly higher in HFD-fed control mice than ND-fed mice. Interestingly, serum leptin levels significantly decreased with spilanthol treatment (Figure 7F). These data suggest that spilanthol suppresses the obesity-induced accumulation of lipids in the liver to improve serum cholesterol content. Spilanthol possibly mediates changes in the hypothalamus to improve energy metabolism and, thus, reduce body weight and leptin levels.

### 3.6. Spilanthol Activates AMPK Signaling to Regulate Lipogenesis and Adipogenesis-Related Transcription Factors in HFD-Induced Obese Mice

To understand whether spilanthol regulates lipid metabolism, hepatic protein expression was measured by Western blot. FAS and pACC protein expression was significantly decreased in spilanthol-treated HFD-fed mice compared to HFD-induced obese mice (Figure 8A). In contrast, ATGL, SIRT1, and AMPK protein expression was significantly higher in spilanthol-treated HFD-fed mice than HFD-induced obese mice (Figure 8B). However, PPARα, PPARγ, C/EBPα, C/EBPβ, and SREBP-1 protein expression was significantly inhibited in spilanthol-treated HFD-fed mice compared to HFD-induced obese mice (Figure 8C). These results suggest that spilanthol inhibited hepatic lipid accumulation via activation of the AMPK pathway, downregulating FAS and ACC expression and suppressing PPARα, PPARγ, C/EBPα, C/EBPβ, and SREBP-1 expression. These results are consistent with previous studies in which activation of AMPK signaling decreased lipogenic and adipogenic transcription factors and promoted FA oxidation [12,13,14]. Therefore, spilanthol may attenuate obesity-induced hepatic lipid accumulation.

## 4. Discussion

Obesity was identified as a chronic disease and can contribute to the development of many chronic diseases, including type 2 diabetes, hyperlipidemia, inflammatory diseases, and cancer. Improving obesity and preventing chronic diseases is the first step to maintaining a healthy lifestyle, and diet is the most important goal [5,6]. Many studies indicate that phyto-compounds from spice plants, including curcumin, capsaicin, and piperine, attenuate chronic metabolic diseases through anti-inflammatory and anti-obesity actions [25,26,27,28]. Compelling evidence supports positive benefits of dietary phyto-compounds for a healthy lifestyle to treat/prevent obesity and chronic metabolic diseases. Therefore, many studies focused on screening phyto-compounds and exploring the molecular mechanisms related to reducing obesity and obesity-related disease. In this study, we investigated the molecular mechanisms underlying the effects of spilanthol, finding that it has anti-obesity effects by mediating AMPK to reduce lipogenic and adipogenic transcriptional factors in 3T3-L1 adipocytes and HFD-induced obese mice.

In obesity, adipocyte hyperplasia and hypertrophy cause excessive accumulation of white adipose tissue (WAT) and increased body weight. Adipogenic transcription factors regulate 3T3-L1 pre-adipocyte differentiation into mature adipocytes [7,8]. Our results indicate that spilanthol significantly suppresses the expression of C/EBP, PPAR, and SREBP-1c expression in the livers of obese mice. In addition, lipogenesis involving FA synthesis and TG synthesis can promote WAT expansion. In the present study, spilanthol suppressed TG synthesis by reducing the expression of FAS and increasing pACC in 3T3-L1 cells and obese mice. The PPARRs also regulate lipogenesis and increase FA β-oxidation in adipose and liver tissue [36]. Here, we found that spilanthol attenuated lipogenesis and adipogenesis exist anti-obesity effect in obese mice mainly by blocking the expression of FAS, ACC, C/EBP, PPAR, and SREBP-1c proteins. SIRT1-mediated AMPK activation increases lipolysis and energy metabolism in obese mice and 3T3-L1 cells [11]. In addition, AMPK suppresses FA synthesis in adipocytes and hepatocytes [37]. Here, we found that spilanthol significantly increased the expression of AMPK, SIRT, ATGL, and pACC-1 proteins, but decreased FAS, in obese mice. We also found that spilanthol may induce AMPK activation to promote CPT1 expression and block lipogenesis in adipocytes. Thus, spilanthol treatment ameliorates obesity with consistent findings in vitro and in vivo.

HFD-induced hyperlipidemia, hyperleptinemia, and increased adipocyte tissue and body weight are major risk factors for metabolic disorders in mice [38]. Previous studies showed that leptin secreted from adipocyte tissue is related to the development of insulin resistance and diabetes [34,35]. Here, spilanthol significantly suppressed body weight gain, reduced the serum levels of leptin and TC, and increased HDL-C, but had no effect on TG levels. In addition, HFD-induced obesity correlated with hepatic TG accumulation [32]. Thus, spilanthol diminished HFD-induced hepatic lipid accumulation in obese mice. We showed that spilanthol significantly inhibits the adipogenic proteins C/EBPα, C/EBPβ, PPARα, PPARγ, and SREBP-1 in the liver, and that spilanthol has notable effects against obesity and associated metabolic abnormalities, such as insulin resistance and hyperlipidemia. Therefore, spilanthol may be an effective agent in the diet for healthy lifestyle management.

It is worth mentioning that spilanthol has high permeation rate from the blood into the brain after intravenous injection in mice [39]. Spilanthol could regulate the central nervous system across the blood–brain barrier [39]. In hypothalamic neurons, increased malonyl-CoA would lead to food intake reduction [40]. In this study, we found that spilanthol inhibited ACC expression. We suggest that spilanthol, through a central action, stimulated energy expenditure but did not reduce appetite.

Furthermore, adipocyte tissue could release leptin to bind to leptin receptors of the hypothalamus to regulating appetite and energy expenditure [41]. In obese individuals, excessive leptin into hypothalamus would increase energy expenditure to decrease body weight [42]. We suggest that the reduced calorie intake (Figure 5D) and serum leptin levels (Figure 7F) with spilanthol produced the anti-obesity effects by promoting energy consumption by increasing the affinity of leptin for its receptor.

Obesity increases the circulation of inflammatory mediators and leptin resistance. Adipocytes release cytokines that induce M2 macrophages to transform into M1 macrophages, and inflammatory M1 macrophages infiltrate the adipose tissue, resulting in more serious chronic inflammatory mediators and insulin resistance in obesity [3,4]. In addition, pre-adipocytes share phenotypic features with macrophages and cause adipose tissue inflammation [31]. Here, we found that LPS-stimulated pre-adipocytes expressed significantly increased levels of COX-2, pJNK, and pp38 proteins. Spilanthol effectively inhibited the expression of COX-2, pJNK, and pp38 proteins but enhanced expression of anti-inflammatory protein HO-1 in LPS-stimulated pre-adipocytes. These findings provide evidence that spilanthol suppresses the levels of COX-2, pJNK, and pp38, which could attenuate or improve inflammatory-induced insulin resistance in adipocytes.

## 5. Conclusions

Our data demonstrate that spilanthol suppresses lipogenesis via activation of AMPK signaling, and inhibits inflammation by blocking JNK and P38 phosphorylation in differentiated 3T3-L1 cells. Moreover, spilanthol attenuates HFD-induced weight gain, fat deposition, and adipocyte size, and alleviates serum TC, HDL-C, and leptin levels. Importantly, spilanthol significantly attenuates obesity-induced hepatic lipid accumulation via activation of AMPK, downregulating lipogenesis and adipogenesis-related transcription factors in obese mice (Figure 9). We conclude that spilanthol is a natural anti-obesity agent that may improve obesity-related metabolic disorders.

## Figures and Tables

**Figure 1 nutrients-11-00991-f001:**
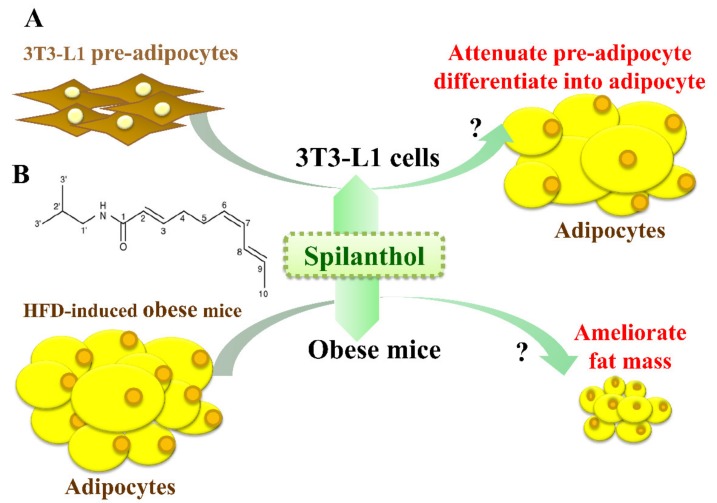
Experimental abstract. (**A**) Study the effects of spilanthol on 3T3-L1 adipocytes and high-fat diet-induced obese mice. (**B**) The structure of spilanthol.

**Figure 2 nutrients-11-00991-f002:**
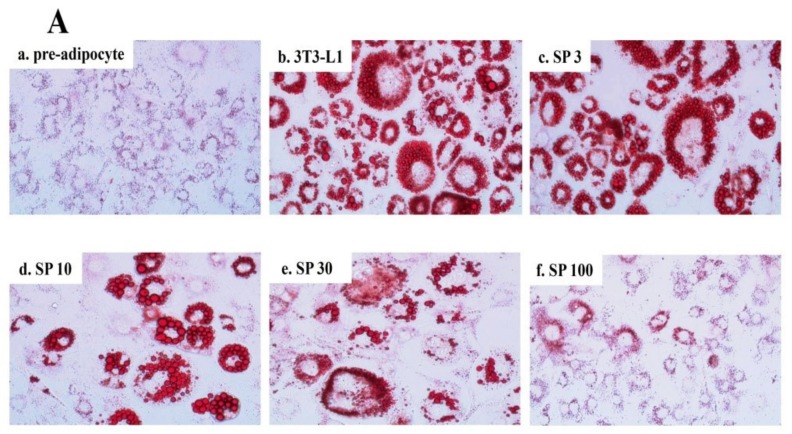
Effect of spilanthol (SP) lipid accumulation and droplets in 3T3-L1 cells. (**A**) 3T3-L1 pre-adipocytes were differentiated in medium containing various concentrations of SP and the differentiated adipocytes examined on day 8 by Oil Red staining: (a) 3T3-L1 pre-adipocytes; (b) 3T3-L1 adipocytes; (c) 3T3-L1 adipocytes + 3 μM SP; (d) 3T3-L1 adipocytes + 10 μM SP; (e) 3T3-L1 adipocytes + 30 μM SP; (f) 3T3-L1 adipocytes + 100 μM SP. (**B**) Intracellular triglyceride accumulation and lipid peroxidation measured using 4′,6-diamidino-2-phenylindole (DAPI) staining.

**Figure 3 nutrients-11-00991-f003:**
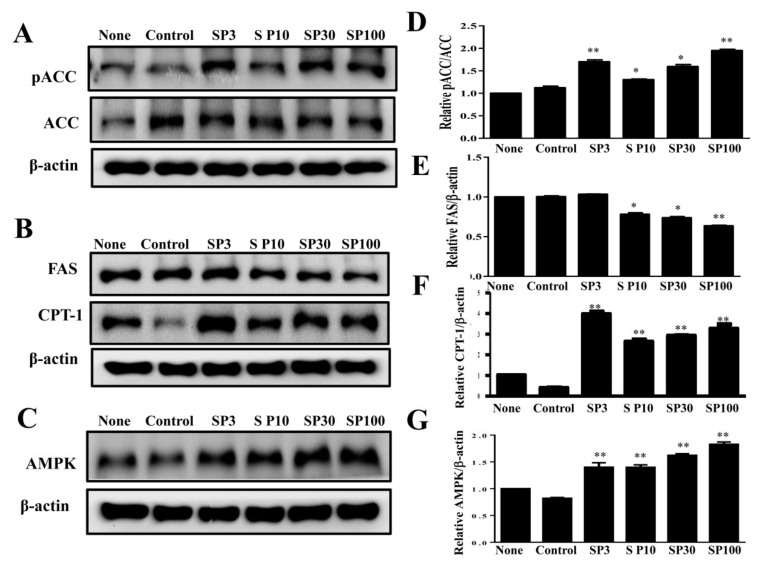
Effects of spilanthol (SP) on lipogenic and AMP-activated protein kinase (AMPK) protein expression in 3T3-L1 adipocytes. Differentiated 3T3-L1 adipocytes (10^4^ cells/mL) were treated with 3–100 μM SP for 24 h. (**A**) Western blots of phosphorylated acetyl-CoA carboxylase (pACC) and ACC, (**B**) fatty-acid synthase (FAS) and carnitine palmitoyltransferase 1 (CPT1), and (**C**) AMPK protein expression. β-Actin was used as an internal control. (**D**) The relative protein levels of pACC/ACC, (**E**) FAS, (**F**) CPT1, and (**G**) AMPK. Data are presented as means ± SD (*n* = 3 per group); * *p* < 0.05, ** *p* < 0.01 compared to differentiated 3T3-L1 cells alone (control group). None: pre-adipocytes; control: differentiated adipocytes; SP3: differentiated adipocytes + 3 μM SP; SP10: differentiated adipocytes + 10 μM SP; SP30: differentiated adipocytes + 30 μM SP; SP100: differentiated adipocytes + 100 μM SP.

**Figure 4 nutrients-11-00991-f004:**
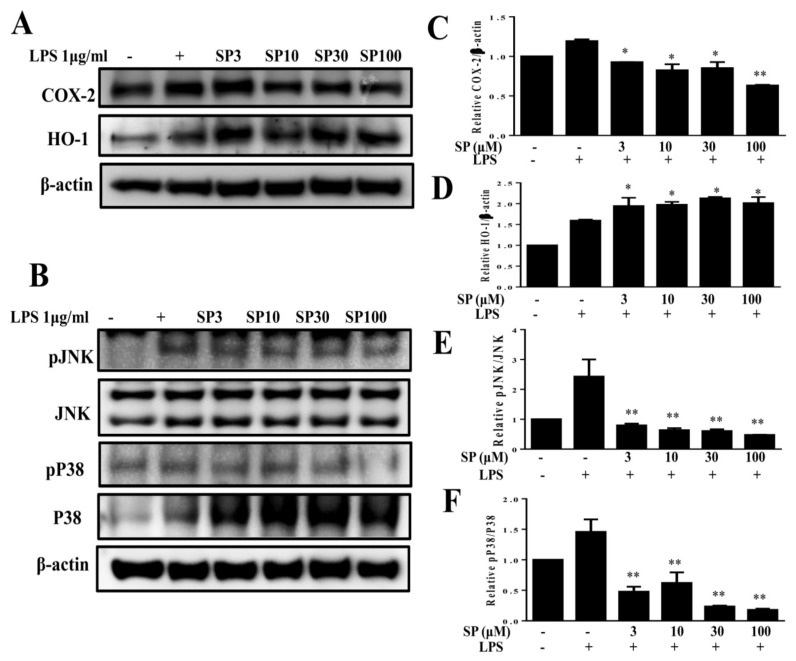
Spilanthol (SP) decreased cyclooxygenase-2 (COX-2), enhanced heme oxygenase-1 (HO-1) protein expression, and inhibited phosphorylation of mitogen-activated protein kinase (MAPK) in LPS-induced 3T3-L1 pre-adipocytes. Here, 10^4^ cells/mL were pretreated with 3–100 μM SP for 1 h and cultured with LPS (1 μg/mL) for 24 h. (**A**) Western blots of COX-2 and HO-1, (**B**) pJNK, JNK, pP33, and P38 proteins (*n* = 3 per group). β-Actin was used as an internal control. (**C**) The relative protein levels of COX-2, (**D**) HO-1, (**E**) pJNK/JNK, and (**F**) pP38/P38. Data are presented as means ± SD; * *p* < 0.05, ** *p* < 0.01 compared to differentiated 3T3-L1 cells alone (control group). None: pre-adipocytes; control: differentiated adipocytes; SP3: differentiated adipocytes + 3 μM SP; SP10: differentiated adipocytes + 10 μM SP; SP30: differentiated adipocytes + 30 μM SP; SP100: differentiated adipocytes + 100 μM SP.

**Figure 5 nutrients-11-00991-f005:**
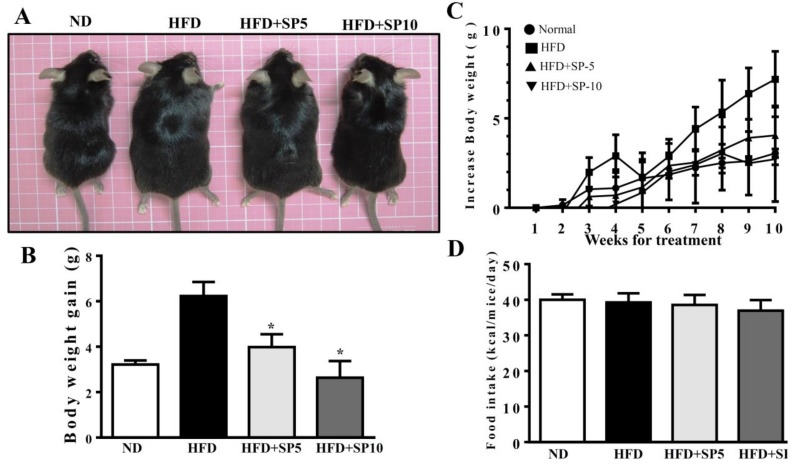
Spilanthol (SP) ameliorates adiposity and decreases body weight gain in high-fat diet (HFD)-induced obese mice. (**A**) Representative images of the whole body. (**B**) Body weight (BW) and (**C**) body weight gain. (**D**) Food intake (kcal/mice/day). C57BL/6 mice were fed a normal diet (ND) or high-fat diet (HFD) with or without a low dose of SP (5 mg spilanthol/kg BW; SP5) or high dose of SP (10 mg spilanthol/kg BW; SP10) for eight weeks. The data are presented as means ± SD, *n* = 8; * *p* < 0.05 compared to HFD-fed mice alone.

**Figure 6 nutrients-11-00991-f006:**
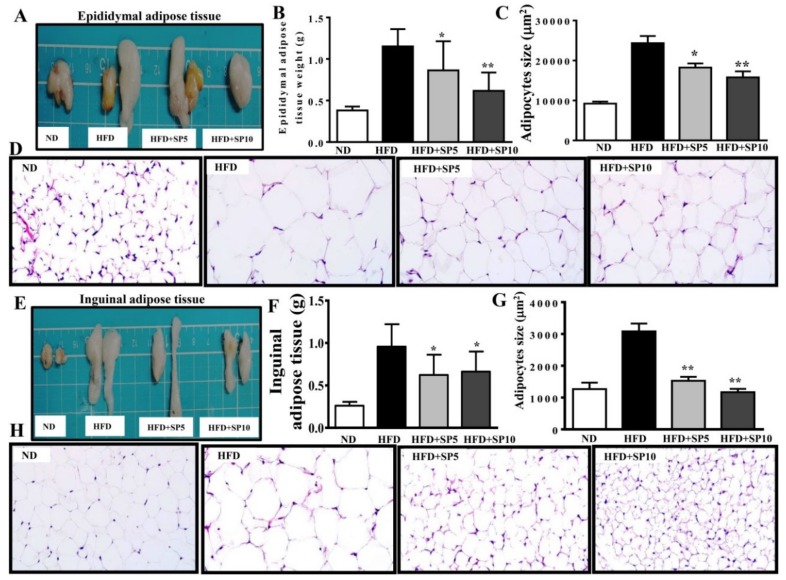
Spilanthol (SP) reduces HFD-induced visceral adipocyte tissue weight and adipocyte size. (**A**) Representative image of epididymal adipose tissue. (**B**) Epididymal adipose tissue weight. (**C**) Epididymal adipocyte size. (**D**) Histological sections of epididymal adipose tissues from mice (400×). (**E**) Representative image of inguinal adipose tissue. (**F**) Inguinal adipose tissue weight. (**G**) Inguinal adipocyte size. (**H**) Histological sections of inguinal adipose tissue (400×). C57BL/6 mice were fed a normal diet (ND) or high-fat diet (HFD) with or without a low dose of SP (5 mg spilanthol/kg BW; SP5) or high dose of SP (10 mg spilanthol/kg BW; SP10) for eight weeks. The data are presented as means ± SD, *n* = 8; * *p* < 0.05, ** *p* < 0.01 compared to HFD-fed mice alone.

**Figure 7 nutrients-11-00991-f007:**
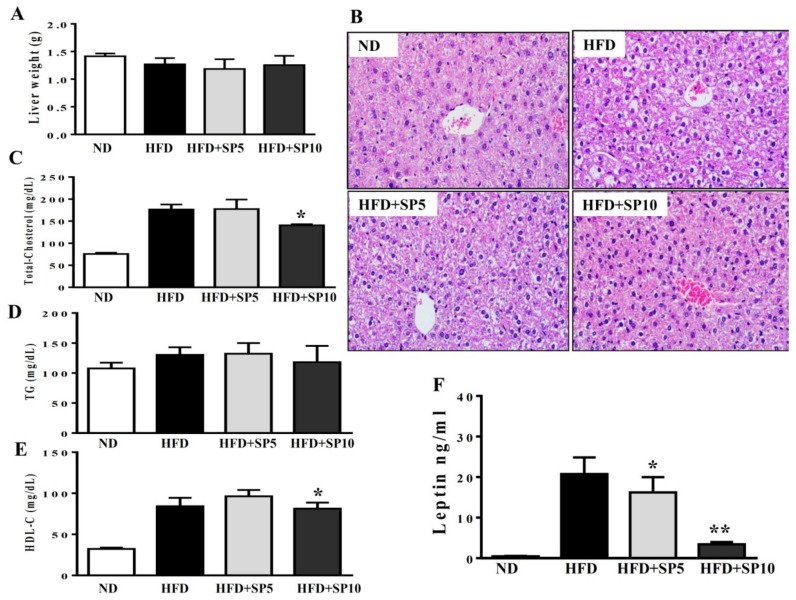
Spilanthol (SP) ameliorates hepatic steatosis, decreases serum leptin levels, and improves serum biochemical parameters in HFD-induced obese mice. (**A**) Final liver weight. (**B**) Hematoxylin and eosin (H&E)-stained liver tissue. (**C**) Serum total cholesterol level. (**D**) Serum triglyceride level. (**E**) Serum high-density lipoprotein cholesterol (HDL-C) level. (**F**) Serum leptin level. C57BL/6 mice were fed a normal diet (ND) or high-fat diet (HFD) with or without a low dose of SP (5 mg spilanthol/kg BW; SP5) or high dose of SP (10 mg spilanthol/kg BW; SP10) for eight weeks. The data are presented as means ± SD, *n* = 8; * *p* < 0.05, ** *p* < 0.01 compared to HFD-fed mice alone.

**Figure 8 nutrients-11-00991-f008:**
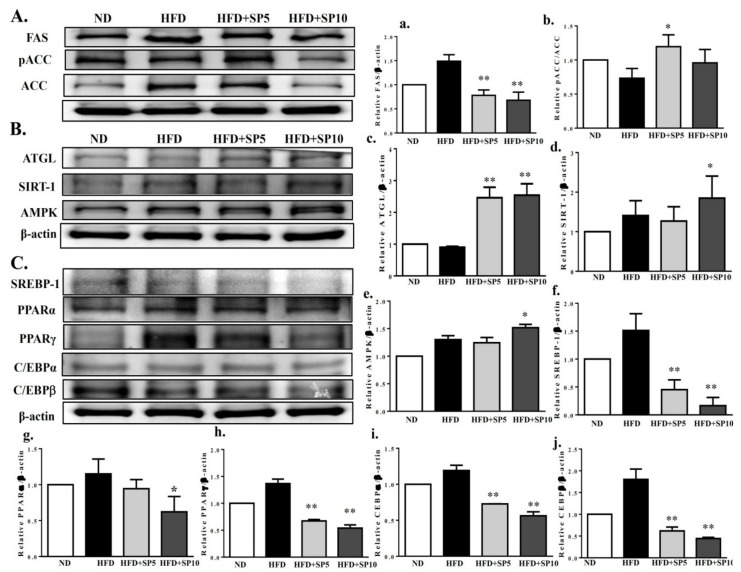
Treatment with spilanthol (SP) downregulates the lipogenesis and adipogenesis-related transcription factors, and upregulates lipolysis-related and AMPK pathway protein expression in the livers of HFD-fed mice. (**A**) Western blots of lipogenesis-related proteins FAS and pACC, (**B**) lipolysis protein triglyceride lipase (ATGL), sirtuin 1 (SIRT1), and AMPK, and (**C**) adipogenic transcription factors sterol regulatory element-binding protein (SREBP-1), peroxisome proliferator-activated receptor (PPAR)α, PPARγ, CCAAT/enhancer-binding protein (C/EBP)α, and C/EBPβ. β-Actin was used as an internal control. The relative protein levels of FAS (a), pACC (b), ATGL (c), SIRT1 (d), AMPK (e), SREBP-1 (f), PPARα (g), PPARγ (h), C/EBPα (i), and C/EBPβ (j) in the liver are also given. Data are presented as means ± SD (*n* = 8); * *p* < 0.05, ** *p* < 0.01 compared to HFD-induced obese mice.

**Figure 9 nutrients-11-00991-f009:**
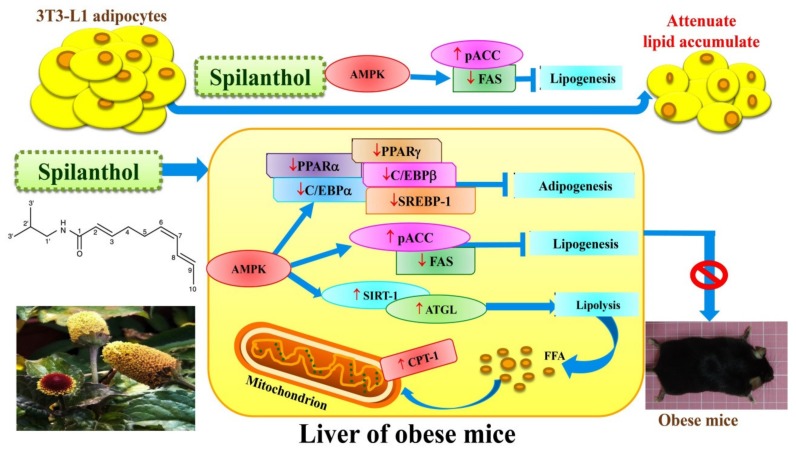
Model explaining the mechanism underlying the anti-obesity effects of spilanthol (SP). SP increases lipolysis-related proteins, suppresses the protein expression of adipogenesis-related transcript factors, and reduces lipogenic proteins via activation of AMPK in the liver of obese mice. SP also inhibits lipogenesis to reduce ACC and FAS expression in 3T3-L1 adipocytes. SP is a potential anti-obesity compound that could ameliorate liver lipid accumulation in obese mice.

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
