# Peer review of "Spilanthol from Traditionally Used Spilanthes acmella Enhances AMPK and Ameliorates Obesity in Mice Fed High-Fat Diet"

_nutrients, 2019, doi:10.3390/nu11050991_

Round 1
Reviewer 1 Report
The authors showed that spilanthol, a bioactive compound in Spilanthes acmella, suppressed lipogenesis in differentiated adipocytes, 3T3-L1 cells and that Spilanthol suppressed weight gain in high-fat diet-fed mice. The authors also indicated several molecular changes in response to spilanthol. The manuscript was very well written and the experiment was well planned and connected. I tend to think that their conclusion and the summary figure are a bit overstating.
Points to be addressed
1 Line 172: 3.1. Spilanthol inhibits lipid accumulation in 3T3‐L1 cells
How was the cytotoxicity of spilanthol? Did the author examine the viability of cells in response to a different dose of spilanthol?
2. Line179: 3.2. Spilanthol regulates lipogenic pathway enzymes in 3T3‐L1 cells
The results in this section are not consistent with the oil Red data in Figure 1. Although I observed an obvious effect only in SP100, most of the SP-treated groups showed protein level changes compared with the control. The author needs to statistically examine whether there is a dose-dependency in Figure 2 results or there are any significant changes among SP groups. If the scheme in Figure 9 is correct, SP3 should reduce lipid in Figure 2A. Otherwise, Figure 9 scheme needs to be modified to make the scheme consistent with the results the authors are presenting.
3. I suggest that the authors mention the possibility that spilanthol can penetrate in the brain and lead to change the energy metabolism through the central action of spilanthol. Also, it is better to discuss that no change in food intake in Figure 5D suggests an increased energy expenditure
Author Response
Response to Reviewer 1 Comments
Nutrients-488509
Title: Spilanthol from Traditionally Used Spilanthes acmella Enhances AMPK and Ameliorates Obesity in Mice Fed High‐Fat Diet
Reviewer #1:
The authors showed that spilanthol, a bioactive compound in Spilanthes acmella, suppressed lipogenesis in differentiated adipocytes, 3T3-L1 cells and that Spilanthol suppressed weight gain in high-fat diet-fed mice. The authors also indicated several molecular changes in response to spilanthol. The manuscript was very well written and the experiment was well planned and connected. I tend to think that their conclusion and the summary figure are a bit overstating.
Points to be addressed
Point 1: Line 172: 3.1. Spilanthol inhibits lipid accumulation in 3T3‐L1 cells. How was the cytotoxicity of spilanthol? Did the author examine the viability of cells in response toa different dose of spilanthol?
Response 1:
Thank for reviewer’s suggestion. The MTT assay was used to evaluate the cytotoxicity of spilanthol in 3T3-L1 adipocytes. Spilanthol at concentrations£ 100 mM showed no significant cytotoxicity in 3T3-L1 cells (data not shown). We added description in 3.1 section [page 5].
Point 2: Line179: 3.2. Spilanthol regulates lipogenic pathway enzymes in 3T3‐L1 cells
The results in this section are not consistent with the oil Red data in Figure 1. Although I observed an obvious effect only in SP100, most of the SP-treated groups showed protein level changes compared with the control. The author needs to statistically examine whether there is a dose-dependency in Figure 2 results or there are any significant changes among SP groups. If the scheme in Figure 9 is correct, SP3 should reduce lipid in Figure 2A. Otherwise, Figure 9 scheme needs to be modified to make the scheme consistent with the results the authors are presenting.
Response 2:
Thank for reviewer’s suggestion.
(1).We according the oil Red data and Fig 3 added discussion description in Discussion section [page 13, line 364].
(2). Taken together Fig 2 and Fig 3, SP³10 mM can inhibited lipogenic and lipid accumulation (P < 0.05 compared with control differentiated adipocytes), obviously. In Fig 3D and 3E, SP=100 mM inhibited lipogenic is the best in all SP-treat groups (P < 0.01 compared to control differentiated adipocytes;P < 0.05 compared to SP-treat groups),showed no statistical difference between SP10 and SP30 mM. Although, SP3 seem to not be significant enough inhibits lipid accumulation in Oil Red data (Fig 2A). However, SP3 can decrease intracellular lipid droplets using of BODIPY493/503 (Fig 2B) and increase pACC to inhibit ACC activity (Fig 3D). In addition, SP significantly decreased liver FAS and pACC protein expression in HFD‐fed mice (Fig 8A). Therefore, we suggested that these results can provide scheme in Figure 9.
(3). We refer to the reviewers ' recommendations to consistent with ours presenting (spilanthol actives AMPK to down-regulate lipogenic pathway) and modified Fig 9 [page 13].
Point 3: I suggest that the authors mention the possibility that spilanthol can penetrate in the brain and lead to change the energy metabolism through the central action of spilanthol. Also, it is better to discuss that no change in food intake in Figure 5D suggests an increased energy expenditure.
Response 3:
Thank for reviewer’s suggestion. We added the possibility that spilanthol can penetrate in the brain and lead to change the energy metabolism through the central action of spilanthol. And discuss that no change in food intake in Figure 5D suggests an increased energy expenditure. Added in the “ Discussion section” line 339 [page 12].

Reviewer 2 Report
This pre-clinical study by Huang et al investigates the in vitro and in vivo effects of Spilanthol (Sp), a plant extract, on adipose tissue metabolism. It was found that in vitro, Sp significantly reduced the expression of a number of proteins involved in lipid synthesis while it increased the expression of those involved in lipid oxidation. In vivo, when administered as intraperitoneal injections to mice fed high-fat diet (HFD), Sp significantly reduced HFD-induced weight gain, visceral adipose tissue expansion and hepatic steatosis associated with increased expression of AMPK over 10 weeks. The effects of Sp were dose-dependent.
Please address the following –
Reduction of leptin level by Sp was described as improvement of leptin resistance [Pg 9]. This description is not appropriate. The reduction in leptin level is likely a consequence/reflection of lower adipose tissue mass (leptin is produced from adipose tissue) until proven otherwise.
In the inguinal and epididymal depots, was there any change in thermogenic genes/proteins expression?
Were other adipose tissue depots such as brown adipose tissue depots examined and what were the findings?
Please describe the rationale for administering Sp twice weekly and for the dosages.
Was glucose tolerance/ insulin resistance assessed?
Figure 5B and 5C both seem to show change in body weight, based on the scale and titles of the y-axes. Please clarify.
Author Response
Response to Reviewer 2 Comments
Nutrients-488509
Title: Spilanthol from Traditionally Used Spilanthes acmella Enhances AMPK and Ameliorates Obesity in Mice Fed High‐Fat Diet
Reviewer 2:
This pre-clinical study by Huang et al investigates the in vitro and in vivo effects of Spilanthol (Sp), a plant extract, on adipose tissue metabolism. It was found that in vitro, Sp significantly reduced the expression of a number of proteins involved in lipid synthesis while it increased the expression of those involved in lipid oxidation. In vivo, when administered as intraperitoneal injections to mice fed high-fat diet (HFD), Sp significantly reduced HFD-induced weight gain, visceral adipose tissue expansion and hepatic steatosis associated with increased expression of AMPK over 10 weeks. The effects of Sp were dose-dependent.
Please address the following –
Point 1:Reduction of leptin level by Sp was described as improvement of leptin resistance [Pg 9]. This description is not appropriate. The reduction in leptin level is likely a consequence/reflection of lower adipose tissue mass (leptin is produced from adipose tissue) until proven otherwise.
Response 1: Thank for reviewer’s suggestion. We corrected the described of SP improvement of leptin resistance [page 9, line 258].
Point 2: In the inguinal and epididymal depots, was there any change in thermogenic genes/proteins expression?
Response 2: Thank for reviewer’s suggestion. We are in the animal experiments about therapeutic effect of spilanthol in anti-obesity effects. This study is our initial stage of the liver lipid metabolism-relative protein expression. We will expect to be more clear therapeutic effects and mechanisms to analysis inguinal and epididymal depots proteins expression, even applied to therapeutic/prevent in obesity-induced muscle atrophy to assay muscle atrophy relative protein and thermogenic genes/proteins expression in future.
Point 3:Were other adipose tissue depots such as brown adipose tissue depots examined and what were the findings?
Response 3: Thank for reviewer’s suggestion. We will expect to be more clear therapeutic effects and mechanisms to analysis spilanthol in brown adipose tissue depots in future. Ours lab have been established obesity-induced skeletal muscle atrophy animal models to explore “run training and intervention spilanthol” improve adipose tissue depots (white adipose tissue class to turn like-brown adipose) and mitochondrion function. According to mouse experiments, except for the extremely cold environment (-4 to -8℃) and “continuous exercise” is a major way lead white adipose tissue class to like-brown adipose, and plant bioactive compound can auxiliary effectiveness. Therefore, this study we primary to explore SP anti-obesity effects (this is the first SP anti-obesity). Then, we can procee obesity-induced skeletal muscle atrophy to explore “run training and intervention spilanthol” to improve adipose tissue depots and mitochondrion function.
Point 4:Please describe the rationale for administering Sp twice weekly and for the dosages.
Response 4: Thank for reviewer’s suggestion. The mice were fed a HFD and received ip 5 mg/kg or 10 mg/kg spilanthol dissolved in DMSO. All of the ip injections were performed twice a week for 10 weeks. At the end of the experimental period, all animals were fasted for 12 h and sacrificed. When we working in pre-experiment using intraperitoneal injection of SP dose =1 and 10 and 20 mg/kg, respectively. The pre-data found that SP=1 mg/kg could not reduce the weight in obese mice. When SP=10 mg/kg and SP=20 mg/kg reduce body weight effect were none different. Hence, we used 5 mg/kg and 10mg/kg as the final two doses in this experiment. In addition, in previous animals experiments we found that daily injections of 50 ul DMSO for a week will affect the appetite of mice, when adjusted two times IP/week does not affect appetite of mice. Therefore, our experiment was designed SP dose =5 mg/kg and 10 mg/kg, using IP (SP) twice weekly.
Point 5:Was glucose tolerance/ insulin resistance assessed?
Response 5: Thank for reviewer’s suggestion. In this study, the postprandial blood glucose was assessed. However, there are none significant statistical differences among all groups (data not shown). We none assessed glucose tolerance/ insulin resistance. In the future, we will evaluate that SP improves diabetes in streptozocin (STZ)-induced diabetic mice. We will assay insulin resistance, HOMA-IR and the molecular mechanisms.
Point 6:Figure 5B and 5C both seem to show change in body weight, based on the scale and titles of the y-axes. Please clarify.
Response 6:
Thank for reviewer’s suggestion. We have modified the scale and titles of the y-axes in Figure 5B and 5C.
